# An Attention-Based Multilayer GRU Model for Multistep-Ahead Short-Term Load Forecasting ^†^

**DOI:** 10.3390/s21051639

**Published:** 2021-02-26

**Authors:** Seungmin Jung, Jihoon Moon, Sungwoo Park, Eenjun Hwang

**Affiliations:** School of Electrical Engineering, Korea University, 145 Anam-ro, Seongbuk-gu, Seoul 02841, Korea; jmkstcom@korea.ac.kr (S.J.); johnny89@korea.ac.kr (J.M.); psw5574@korea.ac.kr (S.P.)

**Keywords:** short-term load forecasting, multistep-ahead forecasting, building electrical energy consumption forecasting, gated recurrent unit, attention mechanism

## Abstract

Recently, multistep-ahead prediction has attracted much attention in electric load forecasting because it can deal with sudden changes in power consumption caused by various events such as fire and heat wave for a day from the present time. On the other hand, recurrent neural networks (RNNs), including long short-term memory and gated recurrent unit (GRU) networks, can reflect the previous point well to predict the current point. Due to this property, they have been widely used for multistep-ahead prediction. The GRU model is simple and easy to implement; however, its prediction performance is limited because it considers all input variables equally. In this paper, we propose a short-term load forecasting model using an attention based GRU to focus more on the crucial variables and demonstrate that this can achieve significant performance improvements, especially when the input sequence of RNN is long. Through extensive experiments, we show that the proposed model outperforms other recent multistep-ahead prediction models in the building-level power consumption forecasting.

## 1. Introduction

Smart grid technologies have attracted much attention because of their potential to cope with climate change and energy crises [1]. A smart grid optimizes energy efficiency through bidirectional interaction between suppliers and consumers and employs renewable energy (i.e., solar and wind power), combined cooling, heating, and power, and energy storage systems (ESSs) [1,2,3]. As a result, greenhouse gases can also be reduced during electricity generation. Short-term load forecasting (STLF) should be performed to determine the power supply required for the power system to establish an operational plan of the smart grid [3,4]. The STLF plays a crucial role in deciding when to run the power generation system from the next hour to one week [4]. Diverse STLF models include daily peak load forecasting, total daily load forecasting, hourly electrical load forecasting, and very STLF (e.g., 10-min and 15-min interval electrical load forecasting). Particularly, hourly electrical load forecasting and very STLF are popularly used for performing demand-side management or operating ESSs from the next hour up to one day [5].

STLF is a challenging task because electrical energy consumption exhibits complicated patterns and fluctuations due to diverse unexpected external factors [6]. Thus, effectively learning the relationship between electrical load and external factors when constructing a forecasting model is essential. Many studies have been done to construct STLF models using artificial intelligence (AI) techniques. For instance, support vector regression (SVR) [7], random forest (RF) [8], and extreme gradient boosting (XGB) [9] have demonstrated excellent prediction performance by considering the nonlinear relationship between input and output variables. However, these STLF models focus on the day-ahead point forecasting; they cannot easily and properly cope with various unexpected events that could affect electrical energy consumption for a day from the present time. Moreover, if the time difference between the training and testing sets is extensive, their prediction performance could deteriorate because they do not adequately reflect recent electrical load trends [10]. 

Recently, a gated recurrent unit (GRU) was used for STLF to solve these problems because it is simple and easy to implement and can carry out multistep-ahead forecasting [11]. However, the GRU has the disadvantage that its forecasting accuracy may deteriorate for a long input sequence because it concentrates on all variables equally. In this paper, we propose a GRU-based multistep-ahead STLF model and augment it using an attention mechanism to improve the forecasting performance by focusing more on crucial variables. To do this, we first collected hourly power consumption data of three office buildings. Then, we carried out preprocessing and exploratory data analysis to verify the relationship between various factors and the power consumption of buildings for model training. Lastly, we constructed an attention-augmented GRU network for multistep-ahead (24 points) forecasting for a day from the current time.

The main contributions of this paper are as follows:We conducted multistep-ahead forecasting for the hourly power consumption of buildings to adequately cope with sudden changes in power consumption caused by various unexpected events, such as peak and blackout, instead of day-ahead point STLF.We constructed an attention-based multilayered GRU model to achieve faster and more stable multistep-ahead STLF than other deep learning (DL) architectures.We verified the superiority of the proposed model through extensive comparisons with several state-of-the-art forecasting models using the power consumption data of three office buildings.

The remainder of this paper is organized as follows. In Section 2, we review several related studies. Section 3 describes the input variable configuration. Section 4 presents the steps for constructing our attention based GRU model. Section 5 shows the experimental results to demonstrate the superiority of the model. Finally, we conclude the study and present the directions for future research in Section 6.

## 2. Related Studies

Many STLF models have been proposed to accurately predict energy consumption based on diverse AI techniques. In this section, we briefly introduce some recent STLF approaches based on machine learning (ML) and DL-based methods and summarize them in Table 1.

Lahouar and Slama [12] proposed a day-ahead hourly electrical load forecasting model based on the RF. They configured input variables by considering two cases: (1) exogenous and relative to the predicted day and (2) endogenous and relative to the previous days. They used the RF model’s variable importance to explain the relationship between the input variables and electrical load. They demonstrated the superiority of the proposed model by comparing it with the persistence analysis, artificial neural network, and SVR. Moon et al. [13] developed an RF-based total daily load forecasting model for university campuses. They first used a moving average method to consider the electrical load patterns on the days of the week. They built an RF model using several input variables, such as the timestamp, temperature, academic year, historical load, and prediction value from the moving average method. They demonstrated that their proposed model outperformed other popular ML methods, such as the decision tree, MLR, gradient boosting machine, SVR, and artificial neural network. Park et al. [14] proposed a two-stage hourly electrical load forecasting model. In the first stage, they constructed two forecasting models using the XGB and RF methods. Then, they combined their prediction values using a sliding window-based multiple linear regression (MLR) model in the second stage. They demonstrated that their proposed model outperforms several single ML methods. 

Ryu et al. [15] proposed two deep neural network-based STLF models for applying a demand-side empirical load database. They compared these STLF models with the shallow neural network, double-seasonal holt-winters, and autoregressive integrated moving average (ARIMA) and demonstrated that their STLF models exhibited excellent performance. Izonin et al. [16] used non-iterative approaches based on a successive geometric transformation model (SGTM) to predict the net hourly electrical energy output of a combined cycle power plant. The SGTM neural-like structure showed a better prediction performance than MLR, SVR, and general regression neural networks. Motepe et al. [17] developed an improved load forecasting process using hybrid AI and deep learning methods. They determined the optimal hyperparameters through several experiments to enhance the performance of long short-term memory (LSTM) networks. They demonstrated that their proposed model, the LSTM network, had lower prediction errors than an adaptive neuro-fuzzy inference system and optimally pruned extreme learning machine. 

Kuan et al. [18] constructed a multilayered self-normalizing GRU model for STLF. They demonstrated that the multilayered self-normalizing technology could improve the prediction performance of the GRU and LSTM models through several experiments. Chitalia et al. [19] presented a short-term load forecasting framework that is robust regardless of building types or locations. They collected five commercial buildings of five different building types located at five different locations. They explored nine different deep learning models and determined the best model for load forecasting in each cluster by considering unsupervised k-mean clustering. Sehovac and Grolinger [20] proposed a recurrent neural network (RNN) with attention for load forecasting. Their RNN had the ability to model time dependencies. They used sequence to sequence (S2S) approach to strengthen the ability by using encoder and decoder. Moreover, they added an attention mechanism to ease the connection between encoder and decoder to further improve the performance. They proved the superiority of their S2S approach and attention mechanism through comparison with a non S2S model and vanilla RNN.

## 3. Data Preprocessing

### 3.1. Data Collection

This section describes the data preprocessing and input variable configuration for the STLF model. For model construction and testing, we used one publicly available dataset and one confidential dataset. The former represents electrical energy consumption data collected from two office buildings in Richland, Washington, USA, from 2 January 2009 to 31 December 2011 [21]. The datasets consist of the timestamp information, temperature in Fahrenheit, and hourly electrical energy consumption. For the three missing values (e.g., 2009/04/05/02:00, 2010/04/04/02:00, and 2011/04/03/02:00) and some anomalous data in the datasets, we handled them properly using a linear interpolation method. The latter represents hourly electrical energy consumption data collected from the headquarters of a midsized company in Seoul, South Korea from 1 January 2015 to 31 December 2017 [10]. We also collected the holiday information in Washington and South Korea at ‘*Time and Date* [22]’ that can confirm national public holiday information in many countries. 

Table 2 lists the simple building information with some statistical analyses on the collected electrical energy consumption data. Holiday includes Saturday, Sunday, and national holidays in the table. The minimum and count in the table represent the lowest electrical energy consumption and the number of data points, respectively. In addition, the first quartile, median, and third quartile represent the values of the lower 25%, 50%, and 75% points, respectively, in the energy consumption. Figure 1 shows the box plots for the energy consumption of three buildings by weekday, holiday, and total.

### 3.2. Feature Extraction

We considered the temperature, timestamp information, and historical load data to build the model and performed feature extraction for the given datasets to train the model as represented in Table 3.

In particular, as electrical energy consumption tends to vary with date and time, we considered all variables that represent date and time such as month, day, hour, day of the week, and holiday. Month, day, and hour have a sequence format. When data with a sequence format are used directly in the AI techniques, it is difficult to reflect their periodicity [23]. For instance, 11 p.m. and midnight are temporally contiguous; however, the data difference in sequence format is 23. 

Hence, we represented the month, day, hour, and day of the week into two-dimensional data using Equations (1) to (8) to reflect their periodicity [23,24]. In Equations (3) and (4), LDM represents the last day of the month. For instance, if the month of the prediction is March, its LDM becomes 31. In addition, we used a variable indicating whether it is a holiday to consider different electrical energy consumption patterns on weekdays and holidays:(1)$${\mathrm{Month}}_{x}=\sin(\left(360{}^{\circ}/12\right)\times \mathrm{Month})$$
(2)$${\mathrm{Month}}_{y}=\cos(\left(360{}^{\circ}/12\right)\times \mathrm{Month})$$
(3)$${\mathrm{Day}}_{x}=\sin(\left(360{}^{\circ}/\mathrm{LDM}\right)\times \mathrm{Day})$$
(4)$${\mathrm{Day}}_{y}=\cos(\left(360{}^{\circ}/\mathrm{LDM}\right)\times \mathrm{Day})$$
(5)$${\mathrm{Hour}}_{x}=\sin(\left(360{}^{\circ}/24\right)\times \mathrm{Hour})$$
(6)$${\mathrm{Hour}}_{y}=\cos(\left(360{}^{\circ}/24\right)\times \mathrm{Hour})$$
(7)$${\mathrm{Day}\;\mathrm{of}\;\mathrm{the}\;\mathrm{week}}_{x}=\sin(\left(360{}^{\circ}/7\right)\times \mathrm{Day}\;\mathrm{of}\;\mathrm{the}\;\mathrm{week})$$
(8)$${\mathrm{Day}\;\mathrm{of}\;\mathrm{the}\;\mathrm{week}}_{y}=\cos(\left(360{}^{\circ}/7\right)\times \mathrm{Day}\;\mathrm{of}\;\mathrm{the}\;\mathrm{week})$$

The temperature, which is closely related to power consumption, is also used for the model training [13]. The datasets of Building 1 and Building 2 included outdoor temperature in Fahrenheit, while the dataset of Building 3 only included power consumption data according to the timestamp information. Hence, we collected the temperature near the Building 3 provided by the Korea Meteorological Administration (KMA) [25] and converted them from Celsius to Fahrenheit.

We also considered the historical electricity load data comprising the electricity load simultaneously from one day to one week ago as the input variables to reflect the recent electricity consumption pattern [26,27]. In addition, we added a new historical electricity load value as an input variable to reflect different electricity load patterns of holidays and weekdays [13]. When the prediction point was a weekday, we used the weekday electricity consumption average of the past week. Likewise, when the prediction point is a holiday, we used the average of the past week’s holiday electricity consumption.

### 3.3. Correlation Analysis

We configured various input variables, such as the timestamp, temperature, and historical load to construct the forecasting model. We calculated the Pearson correlation coefficients with *p*-values for each dataset to confirm their relevance to the actual electrical energy consumption. Table 4 illustrates the results.

In the table, the average load considering the weekdays/holidays, historical load one day before, and historical load one week before exhibit a strong correlation with the actual electrical energy consumption. We also verified that the electric load pattern differs depending on the hour, day of the week, and holiday because Hour_y_, Day of the week_y_, and Holiday present a negative correlation with the actual electrical energy consumption. 

The remaining variables are positively related to the building electrical energy consumption. Although they exhibited a nonlinear relationship, we can adequately handle this by applying a deep learning method, such as the LSTM and GRU networks. Overall, most input variables exhibit a meaningful correlation with the building electrical energy consumption because the p-values are less than 0.01.

## 4. Forecasting Model Construction

### 4.1. Gated Recurrent Unit Model

An artificial neural network (ANN), also known as a multilayer perceptron (MLP), is a popular AI technique implemented based on human biological neurons that can process large amounts of data in parallel and learn efficiently [24,28]. The ANN is a static input-output mapping model that only considers the input and output and does not consider time. It usually represents input, weights, output, etc., in vector form and the output is calculated internally from the input and weight vectors. The decision boundary is orthogonal to the weight vector [29]. In addition, the ANN presents various decision boundaries because an activation function determines the perceptron’s response to its input [23]. 

In contrast, a recurrent neural network (RNN), although it is a kind of ANN, is dynamic input-output mapping model that considers the input of all time. RNNs are well-suited to time-series data because they can process a time-series step-by-step, maintaining an internal state from time step to time step [30]. The LSTM is the most successful and widely used RNN [31]. The LSTM preserves the differential values of old inputs during backpropagation to solve the long-term dependency problem of the RNN. As a variant of the RNN architecture, GRU has the advantage of simplifying the LSTM structure by reducing the computation to update the hidden state while solving the long-term dependency problem and maintaining the performance of the LSTM [11,32].

Figure 2 illustrates a GRU cell architecture. In the figure, the GRU cell has input and forget gates. A gate controller, z, controls both the input and forget gates. When z is 1, the forget gate is closed, and the input gate is open. When z is 0, the forget gate is open, and the input gate is closed. At each step, the previous (t−1) memory is saved, and the input of the time step is cleared. The GRU cell is controlled according to Equations (9)–(12):(9)$$r_{t}=\sigma \left(W_{r}h_{t-1}+U_{r}x_{t}\right)$$
(10)$$z_{t}=\sigma \left(W_{z}h_{t-1}+U_{z}x_{t}\right)$$
(11)$$c_{t}=\tan h\left(W_{c}\left(h_{t-1}⨂r\right)+U_{c}x_{t}\right)$$
(12)$$h_{c}=\left(z⨂c\right)+\left(\left(1-z\right)⨂h_{t-1}\right)$$

We used a GRU network to predict the building electrical energy consumption at 24 points in time (from one hour later to one day later) and considered several hyperparameters to construct our GRU model. We set the number of hidden layers in the GRU model to two. The input layer of the GRU model consists of 18 nodes, and the hidden layer consists of 13 nodes per layer by applying two-thirds of the input layer, plus the size of the output layer [10,23]. 

We used the scaled exponential linear unit (SELU) as an activation function [33,34], defined by Equation (13), where α is a stochastic variable sampled from a uniform distribution at training time and is fixed to 1.67326, which is the expectation value of the distribution at testing time. Moreover, λ is an extra parameter for determining the slope and is set to 1.0507. This is because SELU can effectively train DL models due to its superior self-normalization quality and no vanishing gradient problem [23,34]. 

In addition, we used the Huber loss [35] and adaptive moment estimation (Adam) [34] as a loss function and optimization algorithm, respectively. The Huber loss is calculated using Equation (14) with δ = 1. We set the learning rate and learning epoch to 0.001 and 500, respectively.
(13)$$f\left(\alpha ,\lambda ,x\right)=\{\begin{array}{c} \lambda \left({\alpha e}^{x}-\alpha \right)\;\mathrm{for}\;x<0\\ \lambda x\;\mathrm{for}\;x\geq 0 \end{array}$$
(14)$$L_{\delta }\left(y,f\left(x\right)\right)=\{\begin{array}{c} \frac{1}{2}{\left(y-f\left(x\right)\right)}^{2}\;\mathrm{for}\;\left|y-f\left(x\right)\right|\leq 0\\ \delta \left|y-f\left(x\right)\right|-\frac{1}{2}\;\;\delta^{2}\;\mathrm{otherwise} \end{array}$$

### 4.2. Attention Mechanism

A more extended input sequence in the GRU network results in a worse prediction accuracy of the output sequence because it focuses on all input variables equally, even though they could have different correlations to the forecasting. An attention mechanism can be used to alleviate this problem by focusing on more relevant input variables.

An attention mechanism [36] consists of an encoder that generates an attention vector from the input and a decoder that generates a hidden state by taking the encoder output as input. The hidden states are divided by a view, and the encoder assigns an attention score to the hidden state of each step using the hidden state of the previous view decoder. An attention vector is created by performing a soft-max operation on the created attention score. In this way, whenever the decoder predicts the output value, the encoder concentrates on the input variables similar to the predicted value. 

For instance, Kwon et al. [37] developed an attention mechanism based RNN model on electronic medical records. Hence, we constructed an attention mechanism to focus on input variables with high correlation to improve the model accuracy. We set the size of the attention window to 96. Figure 3 presents the overall attention based GRU model architecture.

## 5. Experimental Results

### 5.1. Experimental Design

In the experiments, we used an Intel (R) Core (TM) i7-8700 CPU (Santa Clara, CA, USA), Samsung 32G DDR4 memory (Suwon, Korea), NVIDIA Geforce GTX 1080ti (Santa Clara, CA, USA) and the operating system is Windows 10 version. For multistep-ahead hourly electricity load forecasting, we performed the experiments in Python 3.7.6, and the RNN-based models were constructed using TensorFlow 1.13.1 [38]. Table 5 presents the entire collection, training set, and testing set period for each building. 

In AI-based STLF model training, normalization is usually conducted to prevent large-scale features from having too much influence by adjusting all the input variables within a close range. For this, we applied the min-max normalization for all input variables using Equation (15). By doing so, we transformed all input variables of the training set and then applied the $X_{\mathrm{norm}}$ values obtained from the training set on the testing set. Hence, the minimum and maximum values of an input variable could be 0 and 1, respectively.
(15)$$X_{\mathrm{norm}}=\frac{X-X_{\min}}{X_{\max}-X_{\min}}$$

We used the mean absolute percentage error (MAPE) and coefficient of variation of the root mean squared error (CVRMSE) to compare the forecasting performance, which can be calculated using Equations (16) and (17), respectively. Both metrics represent the accuracy as a percentage error. Hence, they are more intuitive and easier to understand than other well-known metrics, such as the root mean squared error and mean squared error [23,26]. A lower value for these metrics indicates better prediction performance.
(16)$$\mathrm{MAPE}=\frac{100}{n}\overset{n}{\underset{t=1}{\sum }}\left|\frac{A_{t}-F_{t}}{A_{t}}\right|$$
(17)$$\mathrm{CVRMSE}=\frac{100}{\bar{A}}\sqrt{\frac{\sum_{t=1}^{n}{\left(A_{t}-F_{t}\right)}^{2}}{n}}$$

Here, $\bar{A}$ is an average of the actual values, and $A_{t}$ and $F_{t}$ are the actual and predicted values at time t, respectively.

### 5.2. Experimental Results and Discussion

We compared it with other state-of-the-art models, such as multivariate RF (MRF), DNN, LSTM, ATT-LSTM, GRU, and ensemble models to evaluate the validity of the proposed model. We considered two ensemble models: Park’s stacking ensemble model [14] and Moon’s stacking ensemble model, called COSMOS [10]. We considered all the input variables using each input variable for the prediction time point to construct the MRF and DNN models. Therefore, we used 432 input variables, that is, 18 (number of input variables) × 24 (number of prediction time points), for the multistep-ahead STLF [39]. The two stacking ensemble models consist of two stages, and the second-stage model used the prediction results of the first stage and demonstrated better forecasting performance than many single ML models and existing forecasting models. We also compared the prediction performance of the proposed model with that of Kuan’s GRU model [18]. We implemented an RF-based STLF model using MRF [40] in R packages and the two stacking ensemble models using xgboost 1.3.0 [41] and scikit-learn [42] in the Python environment. The selected hyperparameters for each model are listed in Table 6. 

Table 7, Table 8 and Table 9 and Table 10, Table 11 and Table 12 present the MAPE and CVRMSE comparison of the three buildings by month and date type, respectively. The values in bold font indicate the best performance for the month (or date type). When comparing MAPE (CVRMSE) by month, the MAPE (CVRMSE) values from April to September, which corresponds to the summer, showed the lowest values. When comparing the forecasting performance of weekdays and holidays, the forecasting performance for weekdays was generally better than that for the weekend.

As listed in Table 2, the scale of power consumption of Building 3 is larger than that of Buildings 1 and 2, so the MAPE and CVRMSE values of Building 3 are generally smaller than Buildings 1 and 2. Table 7, Table 8, Table 9, Table 10, Table 11 and Table 12 indicate that the proposed model (ATT-GRU) showed the lowest MAPE among most STLF models in the comparisons by month and date type. In addition, from the results of ATT-LSTM and ATT-GRU, the attention mechanism could improve the prediction performance of the general LSTM and GRU models by 10% or more. 

Although the GRU model performed better than LSTM in most experiments, the GRU model took less time to construct because it was a lighter model than the LSTM model. The average elapsed times for LSTM and GRU-based models are 10,937, 11,581, 7074, and 7263 s for LSTM, ATT-LSTM, GRU, and ATT-GRU, respectively. Also, their training time is depicted in Table 13. In terms of multistep-ahead STLF, the prediction accuracy of the ATT-GRU model did not reveal any significant decrease, even for the rear points. Moreover, our model exhibits the lowest MAPE and CVRMSE values point by point. The point-by-point forecasting results are depicted in the Appendix A. This is because the attention mechanism calculates the weights of the points and focuses on the points with large weights.

Figure 4, Figure 5 and Figure 6 represent scatter plots for the power consumption and temperature of the three buildings. According to the plots, the average power consumption of Buildings 1 and 2 tended to increase similarly as the temperature approached 0 or 100 degrees Fahrenheit and Building 3 showed an increase in power consumption as the temperature approached 100 degrees Fahrenheit. On the contrary, Buildings 1 and 2 showed frequent power consumption peaks when the temperature got close to 0 degrees Fahrenheit. The power forecast accuracy for Buildings 1 and 2 in November and December deteriorated due to unexpected peaks at low temperatures. Our proposed multi-step ahead forecasting method can be used to predict such unexpected peaks and handle them. 

## 6. Conclusions

In this paper, we proposed a reliable hourly electrical energy consumption forecasting scheme using the attention mechanism and GRU network. The GRU network performs multistep-ahead forecasting, and the attention mechanism enables the network to concentrate more on the input variables with a higher correlation to energy consumption. We collected publicly available datasets from the United States Department of Energy and constructed 18 variables related to electricity consumption and an STLF model using the attention based GRU network to verify the proposed scheme performance. Through extensive experiments, we compared the proposed model with other popular state-of-the-art STLF models, including the stacking ensemble model, DNN, COSMOS, LSTM, ATT-LSTM, and GRU. We showed that our model outperformed the other models in terms of MAPE and CVRMSE. In addition, we demonstrated that the prediction performance of LSTM and GRU can be improved by more than 10% by using attention mechanism.

However, our model still has a black box. That is, it cannot explain the evidence of the output like any other deep learning models. To overcome this, we plan to extract feature importance using layer-wise relevance back-propagation and implement explainable AI based on it as a future work. Moreover, we will investigate how further to improve the prediction performance of multistep-ahead electricity load forecasting.

## Figures and Tables

**Figure 1 sensors-21-01639-f001:**
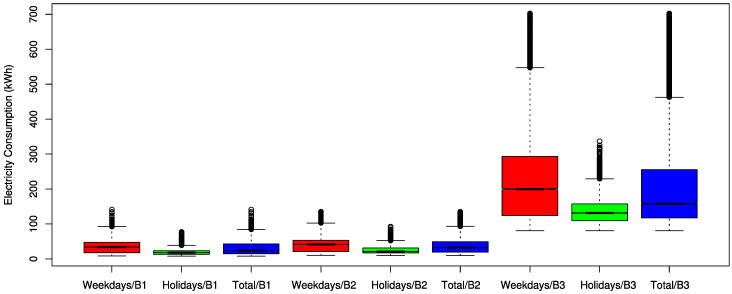
Box plots for the electricity consumptions of three buildings (B1, B2, and B3 represent Building 1, Building 2, and Building 3, respectively).

**Figure 2 sensors-21-01639-f002:**
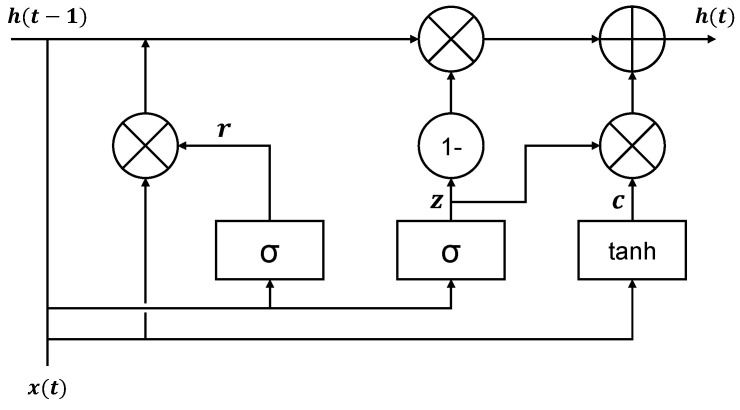
Gated recurrent unit (GRU) cell architecture.

**Figure 3 sensors-21-01639-f003:**
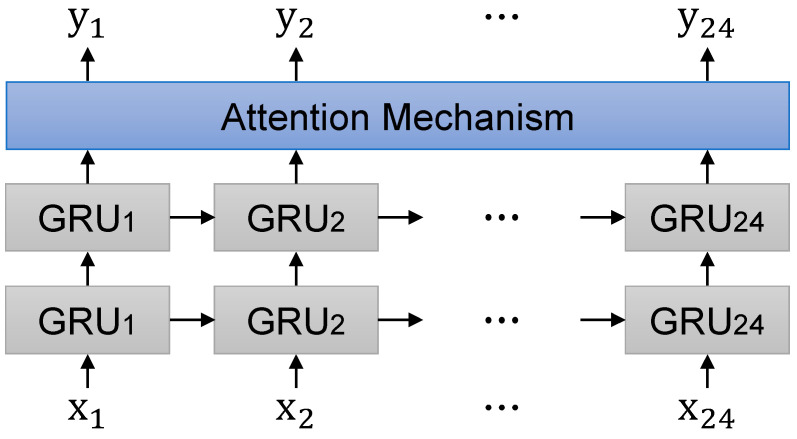
Attention based GRU model architecture.

**Figure 4 sensors-21-01639-f004:**
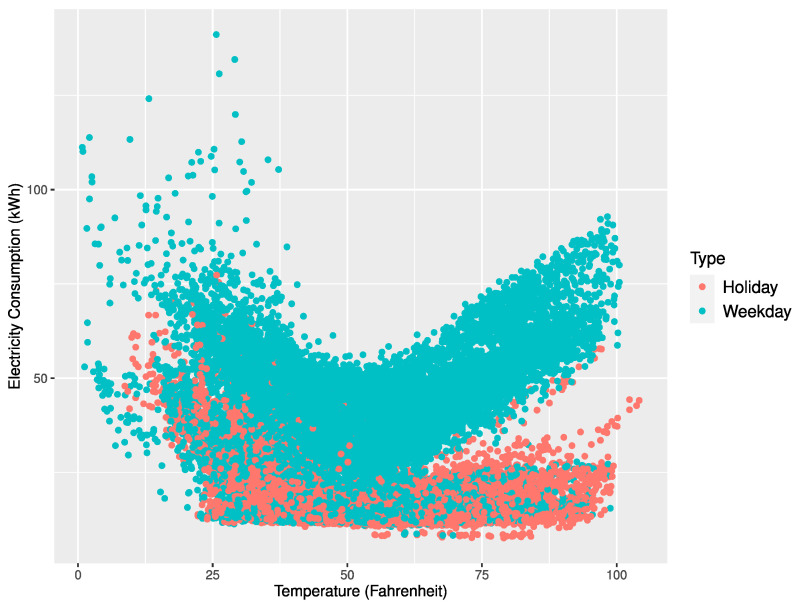
Scatter plot of electricity consumption and temperature for Building 1.

**Figure 5 sensors-21-01639-f005:**
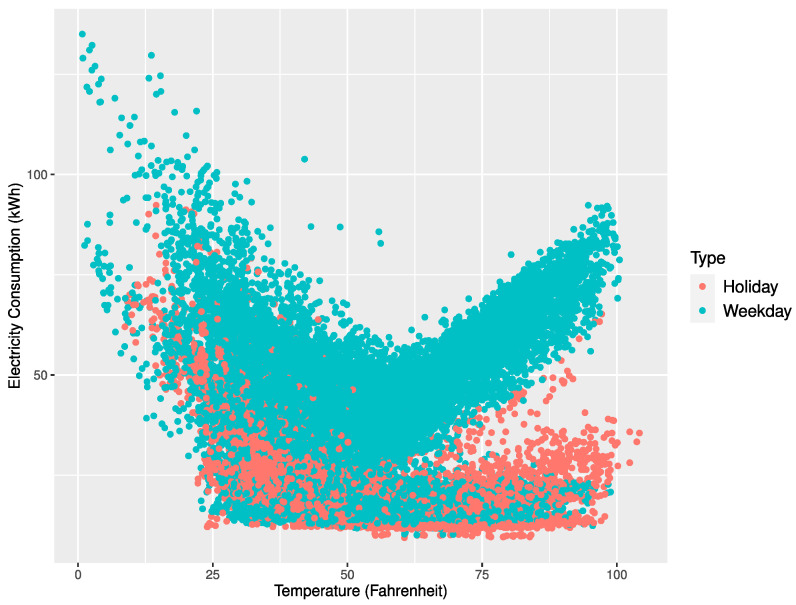
Scatter plot of electricity consumption and temperature for Building 2.

**Figure 6 sensors-21-01639-f006:**
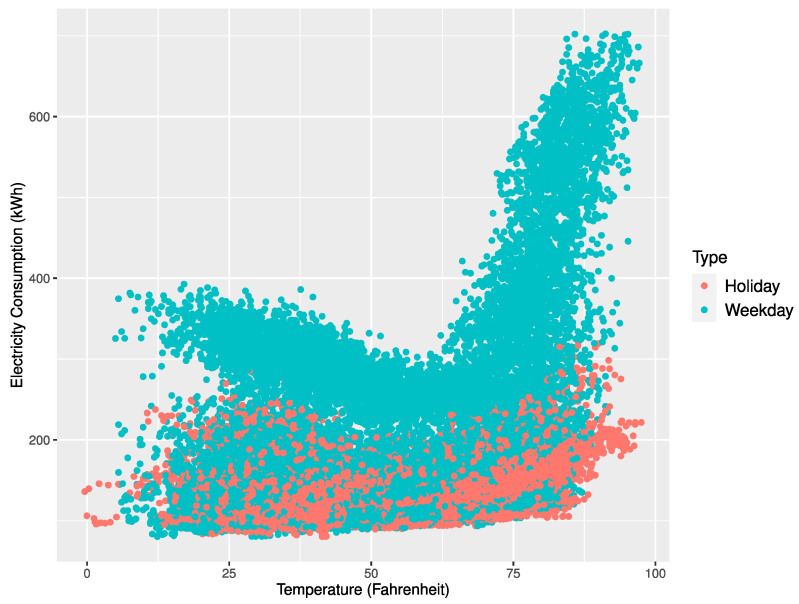
Scatter plot of electricity consumption and temperature for Building 3.

**Table 1 sensors-21-01639-t001:** Summary of related works (RF, random forest; TSCV, time-series cross-validation; XGB, extreme gradient boosting; MLR, multiple linear regression; KEPCO, Korea Electric Power Corporation; DNN, deep neural network, DBN, deep belief network; SBTM, successive geometric transformations model; OP-ELM, optimally pruned extreme learning machines; LSTM, long short-term memory; GRU, gated recurrent unit; BiLSTM, bidirectional long short-term memory; CNN, convolution neural network; S2S, sequence to sequence; RNN, recurrent neural network).

Reference	Datasets	AI Methods	Granularity	Characteristics
Lahouar and Slama [12]	Tunisian Company of Electricity and Gas	RF with TSCV	Hourly	Model interpretation via variable importance
Moon et al. [13]	Private university in Seoul, Korea	RF with TSCV	Daily	Reflecting recent power consumption trends
Park et al. [14]	Commercial and industrial buildings in Korea	Stage 1: RF, XGB Stage 2: MLR with sliding window	Hourly	Performance reinforcement
Ryu et al. [15]	Demand side load data provided by KEPCO	DNN and DBN	Hourly	Multistep-ahead load forecasting
Izonin et al. [16]	Combined cycle power plant dataset	SGTM	Hourly	Less time consuming
Motepe et al. [17]	South African power utility system	OP-ELM and LSTM	30-min	Performance reinforcement
Kuan et al. [18]	Electricity load from Junan, Shanhong province	GRU	15-min	Multistep-ahead load forecasting
Chitalia et al. [19]	Commercial buildings of five different types, in Bangkok, Hyderabad, Virginia, New York, and Massachusetts.	LSTM/BiLSTM, LSTM/BiLSTM with attention, CNN+LSTM, CNN+BiLSTM, and convLSTM/convBiLSTM	15-min	Robust regardless of building types or locations
Sehovac and Grolinger [20]	Commercial building	S2S RNN with attention	5-min	Performance reinforcement

**Table 2 sensors-21-01639-t002:** Energy consumption statistics of three buildings.

Statistical	Building 1			Building 2			Building 3		
	*Weekdays*	*Holidays*	*Total*	*Weekdays*	*Holidays*	*Total*	*Weekdays*	*Holidays*	*Total*
Minimum	8.30	7.60	7.60	10.10	9.40	9.40	80.45	80.47	80.45
1st quartile	17.40	12.90	14.70	20.80	17.00	19.10	124.11	109.53	117.22
Median	33.90	17.70	23.50	41.20	20.90	32.20	199.72	131.14	157.82
Mean	34.43	20.06	29.98	40.10	25.37	35.53	231.46	138.17	201.24
3rd quartile	47.50	23.30	42.50	53.40	31.12	48.80	293.38	157.44	255.41
Maximum	141.10	77.30	141.10	135.00	92.30	135.00	702.53	336.77	702.53
Standard deviation	17.99	9.45	17.18	18.96	12.19	18.45	128.12	35.99	115.86
Count	18,120	8136	26,256	18,120	8136	26,256	17,784	8520	26,304
Location	Richland, Washington			Richland, Washington			Seoul, South Korea		
Public access	Yes			Yes			No		

**Table 3 sensors-21-01639-t003:** List of input variables.

No.	Input Variable	Variable Type
1	Month_x_	Continuous [−1:1]
2	Month_y_	Continuous [−1:1]
3	Day_x_	Continuous [−1:1]
4	Day_y_	Continuous [−1:1]
5	Hour_x_	Continuous [−1:1]
6	Hour_y_	Continuous [−1:1]
7	Day of the week_x_	Continuous [−1:1]
8	Day of the week_y_	Continuous [−1:1]
9	Holiday	Binary [1: holiday, 0: weekday]
10	Temperature	Continuous
11	Historical load_D−7_	Continuous
12	Historical load_D−6_	Continuous
13	Historical load_D−5_	Continuous
14	Historical load_D−4_	Continuous
15	Historical load_D−3_	Continuous
16	Historical load_D−2_	Continuous
17	Historical load_D−1_	Continuous
18	Average load	Continuous

**Table 4 sensors-21-01639-t004:** Pearson correlation coefficients (PCCs) and *p*-values for each dataset.

Input Variable	Building 1		Building 2		Building 3	
	*PCC*	*p-Value*	*PCC*	*p-Value*	*PCC*	*p-Value*
Month_x_	0.042	***	0.118	***	−0.156	***
Month_y_	0.117	***	0.218	***	−0.177	***
Day_x_	0.015	**	0.018	*	−0.009	***
Day_y_	0.002	***	0.008	***	0.001	***
Hour_x_	0.134	***	0.139	***	−0.153	***
Hour_y_	−0.541	***	−0.471	***	−0.640	***
Day of the week_x_	0.333		0.322		0.110	***
Day of the week_y_	−0.048	***	−0.053	***	−0.255	***
Holiday	−0.387	***	−0.369	***	−0.377	***
Temperature	0.014	***	−0.129	***	0.319	***
Historical load_D−7_	0.841	*	0.819	***	0.901	**
Historical load_D−6_	0.657	***	0.642	***	0.726	***
Historical load_D−5_	0.441	***	0.432	***	0.554	**
Historical load_D−4_	0.409	***	0.403	***	0.560	***
Historical load_D−3_	0.424	***	0.421	***	0.564	***
Historical load_D−2_	0.483	***	0.481	***	0.573	***
Historical load_D−1_	0.741	***	0.740	***	0.763	***
Average load	0.903	***	0.885	***	0.959	***

(*p*-values: 0 ‘***’ 0.001 ‘**’ 0.01 ‘*’ 0.05 ‘.’ 0.1 ‘ ’ 1).

**Table 5 sensors-21-01639-t005:** Dataset period configuration.

Dataset Period	Buildings 1 and 2	Building 3
Entire collection period	2009/01/09–2011/12/31	2015/01/08–2017/12/31
Training set	2009/01/09–2010/12/31	2015/01/08–2016/12/31
Testing set	2011/01/01–2011/12/31	2017/01/01–2017/12/31

**Table 6 sensors-21-01639-t006:** Selected hyperparameters for each model.

Models	Package or Module	Selected Hyperparameters
MRF	*MultivariateRandomForest*	No. of trees: 128 [39] No. of features: 144 [39]
DNN	*TensorFlow*	No. of hidden nodes: 289 [23] No. of hidden layers: 5 [23] Activation function: SELU [23] Optimizer: Adam Learning rate: 0.001 No. of epochs: 150 Batch size: 24
Park’s [14] Stage 1: RF, XGB Stage 2: MLR	*scikit-learn* *xgboost*	RF No. of estimators: 256
		XGB No. of estimators: 256 Max depth: 4
		MLR Sliding window size: 24
COSMOS [10] Stage 1: DNNs Stage 2: PCR	*scikit-learn*	DNN No. of hidden nodes: 13 No. of hidden layers: 2, 3, 4, 5 Activation function: ReLU Optimizer: Adam Learning rate: 0.001 No. of epochs: 150 Batch size: 24
		PCR Principal components: 1 Sliding window size: 168
Kuan’s [18]	*TensorFlow*	GRU No. of cells: 100 [18] No. of hidden layers: 5 [18] Activation function: SELU [18] Time step: 12 [18] Batch size: 15 [18]
LSTM ATT-LSTM GRU, ATT-GRU (Ours)	*TensorFlow*	LSTM, GRU No. of hidden nodes: 13 No. of hidden layers: 2 Activation function: SELU Optimizer: Adam Learning rate: 0.001 No. of epochs: 150 Batch size: 24

**Table 7 sensors-21-01639-t007:** MAPE (CVRMSE) comparison by months for Building 1.

Month	MRF	DNN	Park’s [14]	COSMOS [10]	Kuan’s [18]	LSTM	ATT-LSTM	GRU	ATT-GRU
Jan.	21.79 (27.40)	20.36 (27.48)	22.27 (30.62)	13.89 (18.83)	17.34 (26.21)	14.10 (18.31)	13.44 (18.06)	13.86 (18.18)	**13.16 (17.77)**
Feb.	21.32 (29.31)	21.60 (28.55)	18.49 (27.67)	16.46 (20.55)	17.38 (27.93)	16.68 (20.82)	15.67 (20.44)	16.45 (20.55)	**14.51 (19.63)**
Mar.	22.25 (32.89)	22.16 (33.61)	23.24 (32.71)	14.85 (23.00)	18.12 (32.12)	15.24 (22.67)	14.91 **(20.61)**	14.88 (22.57)	**14.66** (21.33)
Apr.	14.24 (19.97)	13.26 (21.43)	20.82 (37.57)	**8.71** (14.62)	11.11 (18.92)	10.72 (16.90)	10.77 (15.81)	10.77 (15.99)	9.12 **(13.37)**
May	7.97 (13.88)	14.67 (22.25)	16.06 (41.02)	6.44 (11.71)	12.13 (20.70)	6.82 (12.97)	7.12 (12.13)	6.47 (13.51)	**6.39 (12.08)**
Jun.	12.54 (13.33)	18.19 (24.70)	20.99 (39.29)	12.56 (12.08)	15.72 (24.11)	12.66 (14.94)	14.14 (18.67)	12.69 (15.48)	**12.18 (14.11)**
Jul.	14.22 (16.42)	16.70 (27.32)	19.21 (35.39)	8.38 (11.37)	16.10 (28.63)	9.04 (16.11)	7.76 (12.51)	8.44 (15.07)	**7.67 (12.02)**
Aug.	14.22 (15.51)	19.40 (28.45)	13.39 (22.53)	7.52 (9.47)	17.30 (28.49)	9.73 (11.11)	8.24 (10.91)	9.44 (11.74)	**7.01 (8.74)**
Sep.	14.20 (15.40)	18.85 (26.70)	28.65 (54.11)	9.66 (13.12)	17.18 (28.55)	10.32 (12.44)	11.41 (13.69)	9.78 (12.54)	**8.96 (11.04)**
Oct.	18.84 (22.80)	15.14 (23.40)	17.94 (28.88)	12.13 (16.54)	17.20 (23.46)	12.40 (14.14)	13.67 (17.76)	12.12 (13.82)	**11.26 (13.51)**
Nov.	31.33 (45.73)	27.22 (42.51)	44.11 (61.05)	24.49 (40.11)	32.57 (39.30)	24.89 (35.82)	20.88 (37.53)	24.47 (35.22)	**20.76 (26.78)**
Dec.	45.11 (47.95)	41.43 (43.57)	41.43 (45.24)	36.64 (38.13)	43.11 (42.18)	37.28 (39.63)	31.76 (41.74)	36.71 (37.88)	**29.12 (31.31)**

**Table 8 sensors-21-01639-t008:** MAPE (CVRMSE) comparison by months for Building 2.

Month	MRF	DNN	Park’s [14]	COSMOS [10]	Kuan’s [18]	LSTM	ATT-LSTM	GRU	ATT-GRU
Jan.	15.81 (18.50)	14.56 (20.43)	13.95 (23.85)	9.10 (11.80)	13.35 (19.99)	8.81 (12.64)	7.64 (10.44)	9.16 (12.12)	**7.16 (9.18)**
Feb.	14.68 (18.83)	17.12 (21.76)	13.05 (20.24)	10.37 (13.99)	14.25 (18.86)	12.73 (15.03)	10.66 (13.61)	10.37 (14.06)	**9.58 (12.27)**
Mar.	12.58 (13.81)	14.60 (17.78)	13.32 (15.77)	**9.02 (11.30)**	14.56 (18.57)	10.86 (9.19)	9.81 (12.46)	9.16 (11.98)	9.09 (11.85)
Apr.	15.82 (16.17)	16.86 (21.08)	20.23 (27.87)	10.60 (13.39)	17.90 (21.28)	12.93 (13.98)	**10.32** (13.38)	10.70 (12.40)	10.88 **(12.56)**
May	17.60 (16.80)	17.62 (25.27)	24.98 (34.07)	7.92 (13.20)	20.04 (24.90)	8.70 (12.22)	7.88 (11.99)	9.08 (12.29)	**7.58 (11.44)**
Jun.	25.31 (18.22)	22.32 (26.79)	33.24 (37.19)	10.31 (11.19)	22.16 (30.44)	10.17 (13.99)	10.12 (13.45)	10.39 (13.91)	**9.56 (12.60)**
Jul.	29.20 (21.19)	27.35 (30.80)	39.94 (46.91)	7.63 (9.57)	28.40 (37.39)	8.82 (8.28)	7.37 (9.67)	8.69 (10.24)	**6.14 (8.15)**
Aug.	28.13 (19.27)	29.73 (30.13)	30.88 (30.37)	7.74 (8.63)	24.06 (34.45)	10.42 (11.48)	7.61 (10.10)	11.31 (13.25)	**6.88 (9.55)**
Sep.	34.78 (23.66)	27.71 (30.75)	43.38 (55.64)	9.56 (11.22)	24.40 (34.78)	9.90 (10.52)	7.96 (10.43)	8.67 (10.63)	**6.97 (9.56)**
Oct.	34.34 (28.69)	19.98 (27.05)	31.51 (34.68)	10.18 (13.62)	21.81 (30.59)	10.23 (9.79)	**8.48 (8.20)**	9.44 (10.27)	8.67 (8.37)
Nov.	36.29 (37.48)	33.78 (37.00)	65.14 (62.40)	21.93 (31.58)	31.25 (48.13)	19.78 (31.98)	19.41 (28.97)	20.75 (30.00)	**18.98 (26.88)**
Dec.	45.07 (39.50)	42.15 (39.32)	38.69 (39.17)	32.50 (34.73)	32.29 (38.27)	32.17 (37.40)	28.80 (36.18)	33.89 (38.71)	**26.77 (33.10)**

**Table 9 sensors-21-01639-t009:** MAPE (CVRMSE) comparison by months for Building 3.

Month	MRF	DNN	Park’s [14]	COSMOS [10]	Kuan’s [18]	LSTM	ATT-LSTM	GRU	ATT-GRU
Jan.	9.02 (12.02)	14.95 (19.29)	9.62 (11.71)	9.38 (11.49)	10.75 (12.37)	9.19 (12.12)	8.71 (11.16)	9.33 (12.41)	**8.05 (10.81)**
Feb.	7.60 (8.94)	13.83 (16.95)	14.74 (33.66)	7.65 (9.18)	9.28 (15.61)	10.02 (13.94)	6.94 (8.32)	9.03 (11.69)	**6.04 (7.66)**
Mar.	5.87 (8.89)	11.28 (15.17)	6.54 (8.93)	4.53 (6.58)	7.52 (10.48)	5.76 (8.44)	**4.86 (7.16)**	7.61 (9.70)	4.95 (7.70)
Apr.	4.28 (5.83)	8.93 (12.83)	5.51 (7.62)	2.99 (4.03)	7.00 (8.93)	4.07 (6.55)	3.17 (5.49)	7.59 (9.29)	**2.38 (3.50)**
May	6.06 (12.31)	11.49 (16.55)	8.26 (14.06)	6.30 (11.95)	7.84 (11.36)	6.85 (8.91)	5.23 (8.71)	7.26 (9.72)	**5.11 (8.45)**
Jun.	7.17 (13.10)	14.76 (23.38)	6.75 (13.84)	5.74 (10.89)	10.05 (17.63)	5.23 (7.19)	5.09 (8.84)	9.77 (11.90)	**4.48 (6.92)**
Jul.	10.18 (15.36)	15.46 (26.38)	9.33 (14.34)	5.89 (8.63)	12.26 (22.42)	5.49 (9.54)	5.63 (8.92)	6.21 (8.14)	**5.33 (8.06)**
Aug.	11.54 (17.38)	16.03 (26.11)	9.51 (15.32)	6.00 (8.95)	10.09 (19.16)	7.14 (9.16)	6.22 (10.21)	7.62 (8.87)	**5.57 (8.48)**
Sep.	6.26 (12.97)	13.33 (19.38)	9.24 (17.82)	5.85 (12.00)	9.26 (15.24)	4.99 (6.19)	4.38 (5.95)	5.10 (6.00)	**3.57 (5.12)**
Oct.	8.08 (13.18)	13.17 (18.16)	7.64 (11.87)	7.97 (14.05)	8.04 (10.92)	7.43 (11.92)	7.51 (10.17)	7.53 (10.55)	**7.20 (9.72)**
Nov.	7.50 (11.15)	13.55 (18.28)	8.76 (13.24)	8.87 (12.67)	9.32 (13.10)	9.10 (14.75)	8.52 (12.72)	9.50 (13.99)	**8.10 (11.20)**
Dec.	10.37 (16.75)	16.39 (23.36)	11.81 (17.74)	9.81 (17.04)	12.88 (19.98)	11.20 (17.59)	9.96 (16.77)	11.02 (17.36)	**9.79 (15.96)**

**Table 10 sensors-21-01639-t010:** MAPE (CVRMSE) comparison by date types for Building 1. WD and HD denote weekdays and holidays, respectively.

Type	MRF	DNN	Park’s [14]	COSMOS [10]	Kuan’s [18]	LSTM	ATT-LSTM	GRU	ATT-GRU
Mon.	18.75 (33.39)	21.78 (33.81)	21.76 (31.80)	13.77 (25.03)	15.47 (29.76)	14.24 (27.35)	13.47 (24.88)	14.15 (26.12)	**12.86 (23.87)**
Tue.	17.57 (25.52)	18.86 (33.85)	32.52 (58.81)	13.08 (21.53)	14.22 (29.76)	13.47 (21.61)	12.31 (20.50)	13.41 (21.41)	**12.06 (20.09)**
Wed.	18.06 (27.77)	21.84 (33.14)	18.63 (33.08)	12.79 (19.81)	17.79 (36.35)	12.56 (19.70)	12.23 (19.12)	12.47 (20.23)	**11.23 (18.40)**
Thu.	18.77 (28.41)	22.08 (30.76)	16.27 (26.51)	12.96 (19.89)	19.17 (38.47)	12.29 (19.36)	10.52 (16.45)	11.75 (18.31)	**9.93 (14.95)**
Fri.	21.20 (29.09)	17.18 (27.93)	17.18 (25.76)	12.66 (23.02)	17.05 (34.21)	12.97 (19.08)	12.66 (18.79)	12.61 (19.51)	**11.43 (18.76)**
Sat.	25.21 (39.61)	21.22 (31.65)	33.97 (45.92)	19.31 (33.02)	13.90 (28.41)	15.03 (28.16)	13.02 **(19.00)**	14.43 (26.45)	**12.41** (19.82)
Sun.	19.30 (29.84)	21.62 (32.29)	26.92 (40.74)	15.57 (29.39)	14.88 (27.88)	16.49 (29.22)	15.48 (28.33)	16.07 (29.54)	**14.66 (27.42)**
WD	18.18 (28.46)	20.16 (31.98)	20.22 (37.28)	12.46 (21.32)	15.61 (31.17)	13.43 (19.10)	12.37 (19.27)	13.06 (19.16)	**11.86 (18.56)**
HD	23.58 (36.96)	21.41 (32.04)	32.20 (45.39)	18.45 (33.35)	14.39 (28.14)	16.01 (28.57)	14.26 (24.30)	15.20 (27.96)	**13.93 (23.69)**

**Table 11 sensors-21-01639-t011:** MAPE (CVRMSE) comparison by date types for Building 2. WD and HD denote weekdays and holidays, respectively.

Type	MRF	DNN	Park’s [14]	COSMOS [10]	Kuan’s [18]	LSTM	ATT-LSTM	GRU	ATT-GRU
Mon.	23.20 (24.79)	24.56 (30.73)	26.13 (28.65)	12.56 (19.32)	17.96 (27.26)	13.59 (20.18)	12.64 (19.15)	13.48 (20.29)	**12.44 (18.28)**
Tue.	22.42 (20.78)	20.60 (25.60)	33.89 (48.70)	11.57 (17.54)	12.57 (21.73)	12.09 (18.19)	11.53 (17.59)	12.41 (18.86)	**11.12 (16.70)**
Wed.	21.02 (20.81)	24.28 (30.05)	24.85 (32.74)	9.66 (14.90)	12.05 (20.06)	10.24 (15.24)	9.24 (14.52)	10.25 (15.28)	**9.01 (13.56)**
Thu.	21.71 (21.49)	26.38 (31.33)	24.63 (25.98)	11.04 (16.95)	16.75 (25.80)	11.07 (16.50)	10.06 (15.89)	11.33 (16.42)	**9.67 (13.32)**
Fri.	25.72 (24.44)	20.61 (25.22)	26.50 (28.33)	11.40 (21.24)	18.92 (29.06)	12.84 (20.86)	10.96 (16.15)	11.85 (19.37)	**10.11 (16.01)**
Sat.	36.37 (44.15)	23.71 (28.44)	43.78 (52.58)	16.51 (28.31)	21.34 (35.08)	17.17 (26.07)	15.60 (25.91)	17.09 (26.47)	**14.58 (25.27)**
Sun.	30.82 (34.81)	24.76 (31.16)	36.05 (42.14)	13.45 (26.88)	22.66 (33.83)	13.91 (23.52)	12.91 (21.09)	13.55 (23.84)	**12.69 (20.56)**
WD	22.12 (21.96)	22.42 (28.97)	25.62 (33.14)	10.53 (17.13)	18.85 (26.94)	12.01 (19.08)	10.45 (17.24)	11.80 (19.14)	**10.15 (16.51)**
HD	34.39 (40.14)	24.55 (30.86)	42.61 (51.59)	16.31 (30.69)	21.80 (33.77)	16.84 (32.83)	15.98 (30.24)	16.55 (31.92)	**13.98 (27.80)**

**Table 12 sensors-21-01639-t012:** MAPE (CVRMSE) comparison by date types for Building 3. WD and HD denote weekdays and holidays, respectively.

Type	MRF	DNN	Park’s [14]	COSMOS [10]	Kuan’s [18]	LSTM	ATT-LSTM	GRU	ATT-GRU
Mon.	7.89 (13.79)	14.35 (23.50)	8.24 (13.39)	6.95 (11.24)	9.65 (14.17)	6.88 (11.61)	6.47 (11.31)	6.74 (11.64)	**6.20 (10.90)**
Tue.	7.60 (8.94)	13.13 (21.35)	9.79 (16.18)	7.08 (11.41)	8.05 (13.17)	7.69 (12.51)	7.42 (12.11)	7.61 (12.40)	**6.79 (11.74)**
Wed.	5.87 (8.89)	13.08 (20.18)	8.85 (17.34)	6.64 (10.23)	6.32 (10.52)	6.93 (10.25)	6.61 (9.83)	6.86 (10.13)	**6.05 (8.69)**
Thu.	4.28 (5.83)	13.87 (23.24)	7.19 (12.32)	5.25 (8.91)	7.47 (13.16)	6.16 (9.97)	4.70 (8.10)	5.82 (9.13)	**4.40 (8.16)**
Fri.	6.06 (12.31)	14.17 (23.64)	10.43 (21.75)	7.78 (12.44)	8.12 (15.97)	7.80 (12.84)	7.01 (12.33)	7.65 (12.67)	**6.54 (11.87)**
Sat.	7.17 (13.10)	12.74 (18.25)	9.27 (13.94)	6.81 (10.23)	8.15 (16.48)	6.45 (10.90)	6.10 (9.39)	6.37 (10.21)	**5.93 (8.87)**
Sun.	10.18 (15.36)	13.72 (22.04)	8.81 (12.14)	6.34 (8.88)	8.08 (16.30)	6.23 (8.58)	5.99 (8.07)	6.08 (8.93)	**5.17 (7.71)**
WD	6.89 (12.16)	13.27 (22.67)	8.48 (16.24)	6.02 (10.30)	8.52 (16.99)	6.60 (10.99)	5.43 (10.25)	6.33 (10.67)	**5.52 (9.84)**
HD	9.79 (19.05)	13.40 (21.09)	9.88 (14.72)	8.07 (12.68)	8.51 (16.46)	8.47 (12.89)	7.78 (12.01)	8.29 (12.70)	**7.95 (11.01)**

**Table 13 sensors-21-01639-t013:** Training time of each model.

Training Time	MRF	DNN	Park’s [14]	COSMOS [10]	Kuan’s [18]	LSTM	ATT-LSTM	GRU	ATT-GRU
Building 1	361	503	274	1345	10,745	10,952	11,581	6974	7108
Building 2	372	498	281	1426	10,769	11,761	12,169	7362	7658
Building 3	364	500	282	1388	10,693	10,097	10,992	6885	7023
Average	367	500	279	1386	10,736	10,937	11,581	7074	7263

## Data Availability

Publicly available datasets were analyzed in this study. This data can be found here: https://openei.org/datasets/dataset/consumption-outdoor-air-temperature-11-commercial-buildings (accessed on 25 February 2021).

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
