# Peer review of "An Attention-Based Multilayer GRU Model for Multistep-Ahead Short-Term Load Forecastingâ€"

_sensors, 2021, doi:10.3390/s21051639_

Round 1
Reviewer 1 Report
This paper considers the problem of short-term electricity forecasting and proposes some new machine-learning techniques (based on a gated recurrent unit network) to make such forecasts. An advantage of the proposed methods is that they try to go beyond statistical learning, in their ability to capture correlations between electricity consumption and influencing factors.
Overall, I found the paper very relevant to the journal, well written, and, as far as I checked, technically sound. The level of innovation is perhaps moderate (in the sense results are not surprising, for such large-scale forecasts), but it is sufficient for publication.
There are, however, some detailed comments and questions the authors should consider and address in their revision, described below.
First, more information about the dataset used is needed. Also, usually in such analysis, data is divided into weekdays and weekends and treating them differently, because the statistical properties of demand data in the two periods are different.
Overall, the author's results show their forecast method works against the state of the art (differences are not too significant, to be honest, but the proposed method is clearly a valuable addition to the mix of available forecasting methods).
Please explain the difference between the proposed methods and Artificial neural networks. Also, why lamda is set to 1.0507 and alpha is fixed to 1.67326.
However, because this is such a large scale model, this reviewer does not find this too surprising, although it is a valuable contribution. It would be more valuable to try to quantify how well the proposed prediction method does for categories of consumers (domestic vs. industrial), for specific time periods (the authors cover the different seasons, which is good, but they could also consider, for example, days when the gird is particularly constrained), and also specific parts of the grid.
In more detail, distribution network operators are usually interested in predicting demand in smaller, more focused areas or communities of energy users, and on particular days, for example very hot summer days when high-demand air-conditioning is used by domestic consumers (i.e. so-called "pinch points" in the system). It would be valuable to say something about this, even if your data does not allow for a full analysis of this case.
Finally, if you could say something about the predictability of the potential for a demand-side response that would be very valuable.
The authors compare the results with their own. Please consider the evaluation of the GRU methods accuracy with ref. 15. I think that 10 self-citation is more for a WOS Journal.
Overall, this is a good paper, if the authors consider the points above.
Author Response
Response to Reviewer 1 Comments
This paper considers the problem of short-term electricity forecasting and proposes some new machine-learning techniques (based on a gated recurrent unit network) to make such forecasts. An advantage of the proposed methods is that they try to go beyond statistical learning, in their ability to capture correlations between electricity consumption and influencing factors. Overall, I found the paper very relevant to the journal, well written, and, as far as I checked, technically sound.
(Response) We really appreciate your kind comments.
The level of innovation is perhaps moderate (in the sense results are not surprising, for such large-scale forecasts), but it is sufficient for publication. There are, however, some detailed comments and questions the authors should consider and address in their revision, described below.
Q1) First, more information about the dataset used is needed. Also, usually in such analysis, data is divided into weekdays and weekends and treating them differently, because the statistical properties of demand data in the two periods are different.
(Response) We put more information about the dataset in Table 1 and added a figure (Figure 1) of box plots on the energy consumptions of each building for weekdays and holidays, and their total.
Table 1. Energy consumption statistics of three buildings.
|
Statistical |
Building 1 |
Building 2 |
Building 3 |
||||||
|
Weekdays |
Holidays |
Total |
Weekdays |
Holidays |
Total |
Weekdays |
Holidays |
Total |
|
|
Minimum |
8.30 |
7.60 |
7.60 |
10.10 |
9.40 |
9.40 |
80.45 |
80.47 |
80.45 |
|
1st quartile |
17.40 |
12.90 |
14.70 |
20.80 |
17.00 |
19.10 |
124.11 |
109.53 |
117.22 |
|
Median |
33.90 |
17.70 |
23.50 |
41.20 |
20.90 |
32.20 |
199.72 |
131.14 |
157.82 |
|
Mean |
34.43 |
20.06 |
29.98 |
40.10 |
25.37 |
35.53 |
231.46 |
138.17 |
201.24 |
|
3rd quartile |
47.50 |
23.30 |
42.50 |
53.40 |
31.12 |
48.80 |
293.38 |
157.44 |
255.41 |
|
Maximum |
141.10 |
77.30 |
141.10 |
135.00 |
92.30 |
135.00 |
702.53 |
336.77 |
702.53 |
|
Standard deviation |
17.99 |
9.45 |
17.18 |
18.96 |
12.19 |
18.45 |
128.12 |
35.99 |
115.86 |
|
Count |
18120 |
8136 |
26256 |
18120 |
8136 |
26256 |
17784 |
8520 |
26304 |
|
Location |
Richland, Washington |
Richland, Washington |
Seoul, South Korea |
||||||
|
Public access |
Yes |
Yes |
No |
||||||
Figure 1. Box plots for the electricity consumptions of three buildings (B1, B2, and B3 represent Building 1, Building 2, and Building 3, respectively).
Q2) Overall, the author's results show their forecast method works against the state of the art (differences are not too significant, to be honest, but the proposed method is clearly a valuable addition to the mix of available forecasting methods). Please explain the difference between the proposed methods and Artificial neural networks.
(Response) In general, ANN refers to the model with problem-solving ability by changing the strength (weight) of synaptic bounding through learning by artificial neurons (nodes). Our proposed model (GRU) is a simplified version of LSTM and can be viewed as a kind of ANN. Unlike general ANN models, GRU model is good at sequence data processing, and hence suitable for processing time series data such as electric load. We added the difference and the related references in Section 4.
“The ANN is a static input-output mapping model that only considers the input and output and does not consider time. It usually represents input, weights, output, etc. in vector form and the output is calculated internally from the input and weight vectors. The decision boundary is orthogonal to the weight vector [30]. In addition, the ANN presents various decision boundaries because an activation function determines the perceptron's response to its input [28].”
“In contrast, a recurrent neural network (RNN), although it is a kind of ANN, is dynamic input-output mapping model that considers the input of all time.”
Q3) Also, why lamda is set to 1.0507 and alpha is fixed to 1.67326.
(Response) These values were first proposed for typical activation normalization in the paper “Self-normalizing neural networks” by G. Klambauer et. al, who designed SELU to solve the exploding and vanishing gradient problem. We added the related reference [33] in the manuscript.
[33] Gunter Klambauer, Tomas Unterthiner, Andreas Mayr, and Sepp Hochreiter. “Self-normalizing neural networks,” arXiv preprint arXiv:1706.02515, 2017.
Q4) However, because this is such a large-scale model, this reviewer does not find this too surprising, although it is a valuable contribution. It would be more valuable to try to quantify how well the proposed prediction method does for categories of consumers (domestic vs. industrial), for specific time periods (the authors cover the different seasons, which is good, but they could also consider, for example, days when the grid is particularly constrained), and also specific parts of the grid.
(Response) According to your comment, we put some tables (Tables 7-12) to analyze the experimental results by month and data type (Days of the Week/Holiday) in Section 5.2, and moved all existing experiment results to the Appendix.
Q5) In more detail, distribution network operators are usually interested in predicting demand in smaller, more focused areas or communities of energy users, and on particular days, for example very hot summer days when high-demand air-conditioning is used by domestic consumers (i.e., so-called "pinch points" in the system). It would be valuable to say something about this, even if your data does not allow for a full analysis of this case.
(Response) Additional analysis on the results are shown in Tables 7-12 as in Q4. In addition, to address particular days, we added scatter plots of the temperature and power consumption on very hot and cold days.
Q6) Finally, if you could say something about the predictability of the potential for a demand-side response that would be very valuable.
(Response) The scatter plot in Q4 shows unexpected peaks in power consumption. This means that such peaks can be predicted and handled by our multi-step ahead forecasting. We put a brief description on this in the manuscript.
Q7) The authors compare the results with their own. Please consider the evaluation of the GRU methods accuracy with ref. 15.
(Response) We added a detailed comparison with the model in the manuscript.
Q8) I think that 10 self-citation is more for a WOS Journal. Overall, this is a good paper, if the authors consider the points above.
(Response) We reduced self-citations a lot according to your suggestion. Thank you.

Reviewer 2 Report
The manuscript deals with a short term load forecasting based on recurrent neural network (LSTM). The manuscript follows standard approach to a scientific reporting, having introduction, related work, proposed approach and evaluation part. While it has a proper structuring, a language is sometimes insufficiently clear or unconcise. Some parts of the manuscripts are not on satisfactory level. With such presentation I found contribution too weak. More precisely:
The abstract does not put strong impression on a reader. Additionally, some of the sentences are confusing, for example:
"However, because these models primarily focus on one-point day- ahead load forecasting, they cannot deal with various unexpected events that could affect the power consumption for a day from the present time." This is simply not true, there are number of methods dealing with one hourly based load prediction.
In several part of iIntroduction (and whole manuscript) I found rather unclear or misleading language. For example "Then, we configure several input variables intimately related to hourly electrical energy consumption". The input variables cannot be configured, these are basically measurements of some process characteristic. In such data driven approach input variables can be only selected or chosen. If the authors are configuring something regarding sensors regarding input variables (e.g. sampling rate, filtering) then it should be clearly stated. However, hereby this is not the case since authors are working with public datasets and manipulating measurements is part of preprocessing step. Therefore I suggest authors to think about every sentence in the manuscript since many of these can be misunderstood due to the language.
The contributions are not properly written in the end of Introduction. For example I don't see any contribution in verification of the proposed method (contribution 3)
The related work is poorly written. The authors should focus only on recent approaches and which are similar (e.g. [10] and others). The authors should focus on dynamical models, STLF (hourly) and residential buildings. Hereby, a critical view is needed, pointing out advantages and disadvantages and making a basis for the proposed approach and latter (sub)sections.
Third section needs some polishing regarding language.
Table 2 in unclear. Variable type contains also the way in which data is preprocessed (so this can be put in third column). Additionally, some variables have x in name, other in subscript. What does x or y stands for? It cannot be clearly seen from Eq. (1) - (8). This section should undergo some heavy editing.
The proposed approach part unresolved questions to the reader. Generally speaking, this must part must undergo significant changes to actually prove reader that such approach can and should be used in particular usecase.
Result section contains some parts which I believe are unnecessary or not written at appropriate place. For example, section 3 contains a list of input variables and then in results section authors are reporting correlation coefficients with respect to load. I found this rather odd since such correlation coefficients are usually driving input variable selection process.
Results section is poorly conducted and written. It does not contain enough details regarding verification. The authors should make more detailed evaluation and discussion regarding results. Apart from that same resultas are presented with tables and graphs which is not necessary.
Author Response
Response to Reviewer 2 Comments
The manuscript deals with a short term load forecasting based on recurrent neural network (LSTM). The manuscript follows standard approach to a scientific reporting, having introduction, related work, proposed approach and evaluation part. While it has a proper structuring, a language is sometimes insufficiently clear or unconcise. Some parts of the manuscripts are not on satisfactory level. With such presentation I found contribution too weak. More precisely:
Q1) The abstract does not put strong impression on a reader. Additionally, some of the sentences are confusing, for example:
"However, because these models primarily focus on one-point day- ahead load forecasting, they cannot deal with various unexpected events that could affect the power consumption for a day from the present time." This is simply not true, there are number of methods dealing with one hourly based load prediction.
(Response) According to your comment, we revised the abstract as follows.
“Abstract: Recently, multistep-ahead prediction has attracted much attention in electric load forecasting because it can deal with various unexpected events that could affect the power consumption for a day from the present time. On the other hand, recurrent neural networks (RNNs), including long short-term memory and gated recurrent unit (GRU) networks, can reflect the previous point well to predict the current point. Due to this property, they have been widely used for multistep-ahead prediction. The GRU model is simple and easy to implement; however, its prediction performance is limited due to the property that it considers all input variables equally. In this paper, we propose a short-term load forecasting model using an attention based GRU to focus more on the crucial variables and demonstrate that this can achieve significant performance improvements, especially when the input sequence of RNN is long. Through extensive experiments, we show that the proposed model outperforms other recent multistep-ahead prediction models in the building-level power consumption forecasting”
Q2) In several part of iIntroduction (and whole manuscript) I found rather unclear or misleading language. For example "Then, we configure several input variables intimately related to hourly electrical energy consumption". The input variables cannot be configured, these are basically measurements of some process characteristic. In such data driven approach input variables can be only selected or chosen. If the authors are configuring something regarding sensors regarding input variables (e.g. sampling rate, filtering) then it should be clearly stated. However, hereby this is not the case since authors are working with public datasets and manipulating measurements is part of preprocessing step. Therefore I suggest authors to think about every sentence in the manuscript since many of these can be misunderstood due to the language.
(Response) According to your suggestion, we fixed the sentence in the Introduction.
“To do this, we first collected hourly power consumption data of three office buildings. Then, we carried out preprocessing and exploratory data analysis to verify the relationship between various factors and the power consumption of buildings for model training.”
Q3) The contributions are not properly written in the end of Introduction. For example, I don't see any contribution in verification of the proposed method (contribution 3)
(Response) We modified the contributions in the Introduction as follows.
- We conducted multistep-ahead forecasting for hourly power consumption of buildings to adequately cope with various unexpected events, such as peak and blackout, instead of day-ahead point STLF.
- We constructed an attention-based multilayered GRU model to achieve faster and more stable multistep-ahead STLF than other deep learning (DL) architectures.
- We verified the superiority of the proposed model through extensive comparisons with several state-of-the-art forecasting models using power consumption data of three office buildings.
Q4) The related work is poorly written. The authors should focus only on recent approaches and which are similar (e.g. [10] and others). The authors should focus on dynamical models, STLF (hourly) and residential buildings. Hereby, a critical view is needed, pointing out advantages and disadvantages and making a basis for the proposed approach and latter (sub)sections.
(Response) We added more related works and their references. We also added Table 1 to summarize these works.
- Izonin, I.; Tkachenko, R.; Kryvinska, N.; Tkachenko, P.; Gregušml, M. Multiple Linear Regression based on Coefficients Identification using Non-Iterative SGTM Neural-Like Structure. In Proceedings of the International Work-Conference on Artificial Neural Networks, Gran Canaria, Spain, 12–14 June 2019; pp. 467–479. https://doi.org/10.1007/978-3-030-20521-8_39
- Chitalia, G.; Pipattanasomporn, M.; Garg, V.; Rahman, S. Robust short-term electrical load forecasting framework for commercial buildings using deep recurrent neural networks. Appl. Energy 2020, 278, 115410. https://doi.org/10.1016/j.apenergy.2020.115410
- Sehovac, L.; Grolinger, K. Deep Learning for Load Forecasting Sequence to Sequence Recurrent Neural Networks With Attention. IEEE Access 2020, 8, 36411–36426. https://doi.org/10.1109/ACCESS.2020.2975738
Table 1. Summary of related works (RF, random forest; TSCV, time-series cross-validation; XGB, extreme gradient boosting; MLR, multiple linear regression; KEPCO, Korea Electric Power Corporation; RBM, restricted Boltzmann machine; SBTM, Successive Geometric Transformations Model; OP-ELM, optimally pruned extreme learning machines; LSTM, long short-term memory; GRU, gated recurrent unit; BiLSTM, bidirectional long short-term memory; CNN, convolution neural network; S2S, sequence to sequence; RNN, Recurrent neural network).
|
Reference |
Datasets |
AI Methods |
Granularity |
Characteristics |
|
Lahouar and Slama [12] |
Tunisian Company of Electricity and Gas |
RF with TSCV |
Hourly |
Explainable, less time consuming |
|
Moon et al. [13] |
Private university in Seoul, Korea |
RF with TSCV |
15-min, hourly, daily |
Reflecting recent power demand trends |
|
Park et al. [14] |
Factory Building in Incheon, General Office Building in Seoul |
Stage 1: RF, XGB Stage 2: MLR with sliding window |
Hourly |
Performance reinforcement, Missing values compensation |
|
Ryu et al. [15] |
Demand side load data provided by KEPCO |
DNN (RBM) |
Hourly |
Multi-step ahead load forecasting |
|
Izonin et al. [16] |
Combined cycle power plant data set |
SGTM |
Hourly |
Less time consuming |
|
Motepe et al. [17] |
South African power utility system |
OP-ELM and LSTM |
30-min |
Performance reinforcement |
|
Kuan et al. [18] |
Electricity loads from Junan, Shanhong province |
GRU |
15-min |
Multi-step ahead forecasting |
|
Chitalia et al. [19] |
Commercial buildings of five different types, in Bangkok, Hyderabad, Virginia, New York, and Massachuset. |
LSTM/BiLSTM, LSTM/BiLSTM with attention, CNN+LSTM, CNN+BiLSTM, and convLSTM/convBiLSTM |
15-min |
Robust regardless of building types or locations |
|
Sehovac and Grolinger [20] |
Commercial building |
S2S RNN with attention |
5-min |
Performance reinforcement |
Q5) Third section needs some polishing regarding language.
(Response) We modified the third section overall.
Q6) Table 2 in unclear. Variable type contains also the way in which data is preprocessed (so this can be put in third column). Additionally, some variables have x in name, other in subscript. What does x or y stands for? It cannot be clearly seen from Eq. (1) - (8). This section should undergo some heavy editing.
(Response) We added more detailed explanation about the table in subsection 3.2.
Q7) The proposed approach part unresolved questions to the reader. Generally speaking, this must part must undergo significant changes to actually prove reader that such approach can and should be used in particular usecase.
(Response) We added two references [19, 20] to show the use case of attention mechanism in load forecasting and one reference [37] to show the use case of attention in time series forecasting of disease. In addition, we appended a brief summarization on the LSTM and RNN models with attention mechanism in related work section.
Ref. 19: “Chitalia, G.; Pipattanasomporn, M.; Garg, V.; Rahman, S. Robust short-term electrical load forecasting framework for com-mercial buildings using deep recurrent neural networks. Appl. Energy 2020, 278, 115410. https://doi.org/10.1016/j.apenergy.2020.115410”
Ref. 20: “Sehovac, L.; Grolinger, K. Deep Learning for Load Forecasting Sequence to Sequence Recurrent Neural Networks With At-tention. IEEE Access 2020, 8, 36411–36426. https://doi.org/10.1109/ACCESS.2020.2975738”
Ref. 37: “Kwon, B.C.; Choi, M.J.; Kim, J.T.; Choi, E.; Kim, Y.B.; Kwon, S.; Sun, J.; Choo, J. Retainvis: Visual analytics with interpretable and interactive recurrent neural networks on electronic medical records. IEEE Trans. Vis. Comput. Graph. 2018, 25, 299–309. https://doi.org/10.1109/TVCG.2018.2865027”
Q8) Result section contains some parts which I believe are unnecessary or not written at appropriate place. For example, section 3 contains a list of input variables and then in results section authors are reporting correlation coefficients with respect to load. I found this rather odd since such correlation coefficients are usually driving input variable selection process.
(Response) According to your suggestion, we reorganized Sections 3 and 5 for more clarity and consistency. Thank you for your suggestion.
Q9) Results section is poorly conducted and written. It does not contain enough details regarding verification. The authors should make more detailed evaluation and discussion regarding results. Apart from that same resultas are presented with tables and graphs which is not necessary.
(Response) For more details regarding verification, we added more tables (Tables 7-12) that contains various comparisons by month and date type. We added detailed evaluation and discussion about the results in lines 318/329.
In addition, we moved the tables for the forecasting results of 24 points to the Appendix and deleted existing graphs.

Reviewer 3 Report
Paper deals with multistep prediction task. Authors proposed an attention-augmented GRU based multistep-ahead STLF model.
Paper has scientific novelty and great practical value.
It has a logical structure. Paper is technically sound. Experimental section is very good.
The proposed approaches are logical, results are clear.
Suggestions:
- Authors should add a Related works section. They should analyse existing ANN-based prediction methods. For example, authors can use non-iterative approaches for solving the stated task. (DOI: 10.1007/978-3-030-20521-8_39 among others)
- Authors should add all optimal parameters for all investigated methods
- It would be good to see the training time for all investigated methods
- Experimental section has a lot of tables and figures. Some of them should be presented in the Appendixes. But I can't find a good Discussion section. Please provide it.
- Conclusion section should be extended using: 1) numerical results obtained in the paper; 2) limitations of the proposed approach; 3) prospects for the future research.
Author Response
Response to Reviewer 3 Comments
Paper deals with multistep prediction task. Authors proposed an attention-augmented GRU based multistep-ahead STLF model. Paper has scientific novelty and great practical value. It has a logical structure. Paper is technically sound. Experimental section is very good. The proposed approaches are logical, results are clear.
(Response) We really appreciate your encouraging comments.
Suggestions:
Q1) Authors should add a Related works section. They should analyze existing ANN-based prediction methods. For example, authors can use non-iterative approaches for solving the stated task. (DOI: 10.1007/978-3-030-20521-8_39 among others)
(Response) We added more related works (references 16, 19, and 20). We also appended Table 1 to summarize the related works.
[16] Izonin, I.; Tkachenko, R.; Kryvinska, N.; Tkachenko, P.; Gregušml, M. Multiple Linear Regression based on Coefficients Identification using Non-Iterative SGTM Neural-Like Structure. In Proceedings of the International Work-Conference on Artificial Neural Networks, Gran Canaria, Spain, 12–14 June 2019; pp. 467–479. https://doi.org/10.1007/978-3-030-20521-8_39
[19] Chitalia, G.; Pipattanasomporn, M.; Garg, V.; Rahman, S. Robust short-term electrical load forecasting framework for commercial buildings using deep recurrent neural networks. Appl. Energy 2020, 278, 115410. https://doi.org/10.1016/j.apenergy.2020.115410
[20] Sehovac, L.; Grolinger, K. Deep Learning for Load Forecasting Sequence to Sequence Recurrent Neural Networks With Attention. IEEE Access 2020, 8, 36411–36426. https://doi.org/10.1109/ACCESS.2020.2975738
Table 1. Summary of related works (RF, random forest; TSCV, time-series cross-validation; XGB, extreme gradient boosting; MLR, multiple linear regression; KEPCO, Korea Electric Power Corporation; RBM, restricted Boltzmann machine; SBTM, Successive Geometric Transformations Model; OP-ELM, optimally pruned extreme learning machines; LSTM, long short-term memory; GRU, gated recurrent unit; BiLSTM, bidirectional long short-term memory; CNN, convolution neural network; S2S, sequence to sequence; RNN, Recurrent neural network).
|
Reference |
Datasets |
AI Methods |
Granularity |
Characteristics |
|
Lahouar and Slama [12] |
Tunisian Company of Electricity and Gas |
RF with TSCV |
Hourly |
Explainable, less time consuming |
|
Moon et al. [13] |
Private university in Seoul, Korea |
RF with TSCV |
15-min, hourly, daily |
Reflecting recent power demand trends |
|
Park et al. [14] |
Factory Building in Incheon, General Office Building in Seoul |
Stage 1: RF, XGB Stage 2: MLR with sliding window |
Hourly |
Performance reinforcement, Missing values compensation |
|
Ryu et al. [15] |
Demand side load data provided by KEPCO |
DNN (RBM) |
Hourly |
Multi-step ahead load forecasting |
|
Izonin et al. [16] |
Combined cycle power plant data set |
SGTM |
Hourly |
Less time consuming |
|
Motepe et al. [17] |
South African power utility system |
OP-ELM and LSTM |
30-min |
Performance reinforcement |
|
Kuan et al. [18] |
Electricity loads from Junan, Shanhong province |
GRU |
15-min |
Multi-step ahead forecasting |
|
Chitalia et al. [19] |
Commercial buildings of five different types, in Bangkok, Hyderabad, Virginia, New York, and Massachuset. |
LSTM/BiLSTM, LSTM/BiLSTM with attention, CNN+LSTM, CNN+BiLSTM, and convLSTM/convBiLSTM |
15-min |
Robust regardless of building types or locations |
|
Sehovac and Grolinger [20] |
Commercial building |
S2S RNN with attention |
5-min |
Performance reinforcement |
Q2) Authors should add all optimal parameters for all investigated methods.
(Response) We added Table 6 to show all optimal parameters for each model.
Table 6. Models and their optimal hyperparameters.
|
Models |
Package or Module |
Selected Hyperparameters |
|
MRF |
MultivariateRandomForest |
No. of trees: 128 [39] No. of features: 144 [39] |
|
DNN |
TensorFlow |
No. of hidden nodes: 289 [23] No. of hidden layers: 5 [23] Activation function: SELU [23] Optimizer: Adam Learning rate: 0.001 No. of epochs: 150 Batch size: 24 |
|
Park’s [14] Stage 1: RF, XGB Stage 2: MLR |
scikit-learn xgboost |
RF No. of estimators: 256 |
|
XGB No. of estimators: 256 Max depth: 4 |
||
|
MLR Sliding window size: 24 |
||
|
COSMOS [10] Stage 1: DNNs Stage 2: PCR |
scikit-learn |
DNN No. of hidden nodes: 13 No. of hidden layers: 2, 3, 4, 5 Activation function: ReLU Optimizer: Adam Learning rate: 0.001 No. of epochs: 150 Batch size: 24 |
|
PCR Principal components: 1 Sliding window size: 168 |
||
|
Kuan’s [18] |
TensorFlow |
GRU No. of cells: 100 [18] No. of hidden layers: 5 [18] Activation function: SELU [18] Time step: 12 [18] Batch size: 15 [18] |
|
LSTM ATT-LSTM GRU, ATT-GRU (Ours) |
TensorFlow |
LSTM, GRU No. of hidden nodes: 13 No. of hidden layers: 2 Activation function: SELU Optimizer: Adam Learning rate: 0.001 No. of epochs: 150 Batch size: 24 |
Q3) It would be good to see the training time for all investigated methods.
(Response) We appended a brief description on the computer environment in Section 5.2 and the training time for all models in Table 13.
Table 13. Training time of each model.
|
Training time |
MRF |
DNN |
Park’s [14] |
COSMOS [10] |
Kuan’s [18] |
LSTM |
ATT-LSTM |
GRU |
ATT-GRU |
|
Building 1 |
361 |
503 |
274 |
1345 |
10745 |
10952 |
11581 |
6974 |
7108 |
|
Building 2 |
372 |
498 |
281 |
1426 |
10769 |
11761 |
12169 |
7362 |
7658 |
|
Building 3 |
364 |
500 |
282 |
1388 |
10693 |
10097 |
10992 |
6885 |
7023 |
Q4) Experimental section has a lot of tables and figures. Some of them should be presented in the Appendixes. But I can't find a good Discussion section. Please provide it.
(Response) We moved the experimental results to the Appendix and deleted graphs. Instead, we put new tables (Tables 7-12) in Section 5.2 to show the experimental results by the month and data types (Days of the Week/Holiday) and added a brief discussion on the results.
We also added the scatter plots (Figure 3-5) and their interpretation.
Q5) Conclusion section should be extended using: 1) numerical results obtained in the paper; 2) limitations of the proposed approach; 3) prospects for the future research.
(Response) According to your comment, we revised the conclusion.
“We showed that our model outperformed the other models in terms of MAPE and CVRMSE. In addition, we demonstrated that the prediction performance of LSTM and GRU can be improved more than 10% by using attention mechanism.”
“However, our model still has a black box. That is, it cannot explain the evidence of the output like any other deep learning models. To overcome this, we plan to extract feature importance using layer-wise relevance back-propagation and implement explainable AI based on it as a future work.”

Round 2
Reviewer 1 Report
The reviewer has no further comments, authors have satisfactorily addressed most of my comments.
Author Response
Response to Reviewer 1 Comments
The reviewer has no further comments, authors have satisfactorily addressed most of my comments.
(Response) We really appreciate your kind comments.

Reviewer 2 Report
Authors resolved all the issues that I raised in my previous review. I have only several objections regarding usage of the term "unexpected events" in Abstract and in contribution number 1.
Authors should reconsider to change first sentence of Abstract.
Basically, models which provide hourly based prediction simply provide information at different sampling rate and basically capture dynamics of the process in more details. They can not deal with unexpected events, they can just point out that something unexpected happened.
Author Response
Response to Reviewer 2 Comments
Q) Authors resolved all the issues that I raised in my previous review. I have only several objections regarding usage of the term "unexpected events" in Abstract and in contribution number 1.
Authors should reconsider to change first sentence of Abstract.
Basically, models which provide hourly based prediction simply provide information at different sampling rate and basically capture dynamics of the process in more details. They can not deal with unexpected events, they can just point out that something unexpected happened.
(Response) According to your comment, we revised the first sentence in the abstract.
From: “Recently, multistep-ahead prediction has attracted much attention in electric load forecasting because it can deal with various unexpected events that could affect the power consumption for a day from the present time.”
To: “Recently, multistep-ahead prediction has attracted much attention in electric load forecasting because it can deal with sudden changes in power consumption caused by various events such as fire and heatwave for a day from the present time.”
We also revised the contribution number 1.
From: “We conducted multistep-ahead forecasting for hourly power consumption of buildings to adequately cope with various unexpected events, such as peak and blackout, instead of day-ahead point STLF.”
To: “We conducted multistep-ahead forecasting for hourly power consumption of buildings to adequately cope with sudden changes in power consumption caused by various unexpected events, such as peak and blackout, instead of day-ahead point STLF.”

Reviewer 3 Report
Authors did a lot of work. Paper can be accepted!
Author Response
Response to Reviewer 3 Comments
Authors did a lot of work. Paper can be accepted!
(Response) We really appreciate your encouraging comments.
